# Structural insights into heterohexameric assembly of epilepsy-related ligand–receptor complex LGI1–ADAM22

**Takayuki Yamaguchi[1†], Kei Okatsu[1†], Masato Kubota[1], Ayuka Mitsumori[1], Atsushi Yamagata[2], Yuko Fukata[3], Masaki Fukata[4], Mikihiro Shibata[5,6*], Shuya Fukai[1*]**

[1]Department of Chemistry, Graduate School of Science, Kyoto University, Kyoto, Japan; [2]Laboratory for Protein Functional and Structural Biology, RIKEN Center for Integrative Medical Sciences, Kanagawa, Japan; [3]Division of Molecular and Cellular Pharmacology, Nagoya University Graduate School of Medicine, Nagoya, Japan; [4]Division of Neuropharmacology, Nagoya University Graduate School of Medicine, Nagoya, Japan; [5]Institute for Frontier Science Initiative, Kanazawa University, Kanazawa, Japan; [6]WPI Nano Life Science Institute (WPI-NanoLSI), Kanazawa University, Kanazawa, Japan

**\*For correspondence:**
msshibata@staff.kanazawa-u.ac.jp (MS);
fukai@kuchem.kyoto-u.ac.jp (SF)

[†]These authors contributed equally to this work

**Competing interest:** The authors declare that no competing interests exist.

## eLife Assessment

In this **convincing** work by Yamaguchi et al. the cryo-EM structure of the heterohexameric 3:3 LGI1-ADAM22 complex is presented. The findings suggest that LGI1 can cluster ADAM22 in a trimeric fashion. The clustering of cell surface proteins is **important** in controlling signaling in the nervous system. This new version of the manuscript has been improved substantially and the figures have been enhanced and clarified.

**Abstract** Leucine-rich glioma-inactivated 1 protein (LGI1) is a secreted neuronal protein consisting of the N-terminal leucine-rich repeat (LRR) and C-terminal epitempin-repeat (EPTP) domains. LGI1 is linked to epilepsy, a neurological disorder that can be caused by genetic mutations of genes regulating neuronal excitability (e.g. voltage- or ligand-gated ion channels). ADAM22 is a membrane receptor that binds to LGI1 extracellularly and interacts with AMPA-type glutamate receptors via PSD-95 intracellularly to maintain normal synaptic signal transmission. Structural analysis of the LGI1–ADAM22 complex is important for understanding the molecular mechanism of epileptogenesis and developing new therapies against epilepsy. We previously reported the crystal structure of a 2:2 complex consisting of two molecules of LGI1 and two molecules of the ADAM22 ectodomain (ECD), which is suggested to bridge neurons across the synaptic cleft. On the other hand, multiangle light scattering, small-angle X-ray scattering, and cryo-electron microscopy (cryo-EM) analyses have suggested the existence of a 3:3 complex consisting of three molecules of LGI1 and three molecules of ADAM22. In the previous cryo-EM analysis, many observed particles were in a dissociated state, making it difficult to determine the three-dimensional (3D) structure of the 3:3 complex. In this study, we stabilized the 3:3 LGI1–ADAM22$_{ECD}$ complex using chemical cross-linking and determined the cryo-EM structures of the LGI1$_{LRR}$–LGI1$_{EPTP}$–ADAM22$_{ECD}$ and 3:3 LGI1–ADAM22$_{ECD}$ complexes at 2.78 Å and 3.79 Å resolutions, respectively. Furthermore, high-speed atomic force microscopy (HS-AFM) visualized the structural features and flexibility of the 3:3 LGI1–ADAM22$_{ECD}$ complex in solution. We discuss new insights into the interaction modes of the LGI1–ADAM22 higher-order complex and the structural properties of the 3:3 LGI1–ADAM22 complex.

## Introduction

Epilepsy is a prevalent neurological disorder, affecting approximately 1% of the population. Epilepsy is characterized by recurrent, unprovoked seizures, resulting from an imbalance between excitation and inhibition within neural circuits. Mutations associated with epilepsy frequently occur in genes that regulate neuronal excitability through ion channels such as voltage-gated ion channels (e.g. $K^+$, $Na^+$, and $Ca^{2+}$ channels) and ligand-gated ion channels (e.g. nicotinic acetylcholine and $GABA_A$ receptors) (*Fukata and Fukata, 2017*; *Noebels, 2015*; *Steinlein, 2004*). Additionally, some epilepsy-related mutations have been identified in genes encoding non-ion channel proteins such as *LGI1* (*Gu et al., 2002*; *Kalachikov et al., 2002*; *Morante-Redolat et al., 2002*; *Staub et al., 2002*).

LGI1 is a 60 kDa secreted neuronal protein that consists of the N-terminal leucine-rich repeat (LRR) domain and the C-terminal epitempin-repeat (EPTP) domain (*Staub et al., 2002*; *Figure 1a*). Mutations in the LGI1 gene, resulting in incorrect folding and posttranslational modifications, cause autosomal dominant epilepsy with auditorial features (ADEAF) (*Gu et al., 2002*; *Kalachikov et al., 2002*; *Morante-Redolat et al., 2002*). For example, the E383A mutant of LGI1 is eliminated by the endoplasmic reticulum and becomes deficient in secretion (*Yokoi et al., 2015*). Meanwhile, the S473L mutation of LGI1 causes epileptiform seizures due to reduced binding to ADAM22, which is a member of the A disintegrin and metalloproteinase (ADAM) family (*Fukata et al., 2010*; *Schulte et al., 2006*) and acts as a receptor for LGI1 in neurons without protease activity (*Fukata et al., 2006*; *Sagane et al., 1999*). ADAM22 is a single-pass transmembrane protein. The ectodomain (ECD) of ADAM22 consists of a metalloprotease-like domain, a disintegrin domain, a cysteine-rich domain, and an EGF-like domain (*Liu et al., 2009*; *Figure 1a*). The metalloprotease-like domain interacts with the EPTP domain of LGI1 in the extracellular space (*Fukata et al., 2006*; *Yamagata et al., 2018*). In the cytoplasm, the PDZ-binding motif-containing C-terminal tail of ADAM22 binds to a synaptic scaffolding protein PSD-95, which regulates the cellular dynamics of α-amino-3-hydroxy-5-methyl-4-isoxazolepropionic acid (AMPA) receptors through binding to Stargazin, an auxiliary subunit of AMPA receptors, in the postsynapse (*Chen et al., 2000*; *Nicoll et al., 2006*). Furthermore, LGI1 forms a complex with the voltage-gated potassium channel through ADAM22/23 (*Fukata et al., 2021*; *Schulte et al., 2006*; *Seagar et al., 2017*). As such, the LGI1–ADAM22 complex maintains normal nerve signal transduction through the interaction network, including synaptic ion channels (*Fukata et al., 2006*; *Fukata et al., 2010*; *Lovero et al., 2015*). Structural analysis of the LGI1–ADAM22 complex is important for understanding the molecular basis of epileptogenesis and for developing therapeutic strategies based on this understanding.

We previously reported the crystal structure of a 2:2 LGI1–ADAM22ECD complex consisting of two molecules of LGI1 and two molecules of ADAM22ECD (*Yamagata et al., 2018*). The long-axis length of the 2:2 complex is approximately 190 Å, which is comparable to the width of the synaptic cleft. This 2:2 complex structure and the structure-guided study on a mouse model for familial epilepsy suggested that the formation of the 2:2 complex bridges neurons in the synaptic cleft. The results revealed the structural basis of the interaction between the EPTP domain of one LGI1 and the LRR domain of the other LGI1, as well as the interaction between the EPTP domain of LGI1 and the metalloproteinase-like domain of ADAM22 (*Yamagata et al., 2018*). On the other hand, size-exclusion chromatography-multiangle light scattering (SEC-MALS), size-exclusion chromatography-small-angle X-ray scattering (SEC-SAXS), and cryo-electron microscopy (cryo-EM) analyses suggested that three molecules of LGI1 and three molecules of ADAM22ECD bind to each other to form a 3:3 complex at near physiological salt concentrations (*Yamagata et al., 2018*). Similarly to the 2:2 complex, the 3:3 complex might serve as an extracellular scaffold to stabilize Kv1 channels or AMPA receptors in a *trans*-synaptic fashion (*Fukata et al., 2021*; *Lovero et al., 2015*; *Schulte et al., 2006*). In addition, the 3:3 assembly in a *cis* fashion on the same membrane might regulate the accumulation of Kv1 channel complexes at the axon initial segment (*Hivert et al., 2019*; *Seagar et al., 2017*). However, no clear evidence to prove these potential mechanistic roles of the 3:3 assembly has been provided, and the 3D structure of the 3:3 complex has not yet been determined.

In this study, the 3D structure of the 3:3 LGI1–ADAM22ECD complex was determined at a nominal resolution of 3.79 Å by cryo-EM single-particle analysis. We also determined the 3D structure of the LGI1LRR–LGI1EPTP–ADAM22ECD complex at a nominal resolution of 2.78 Å. These higher-resolution 3D structures provide more detailed interaction mechanisms for the LGI1–ADAM22 higher-order assembly. We also performed high-speed atomic force microscopy (HS-AFM) (*Ando et al., 2001*;

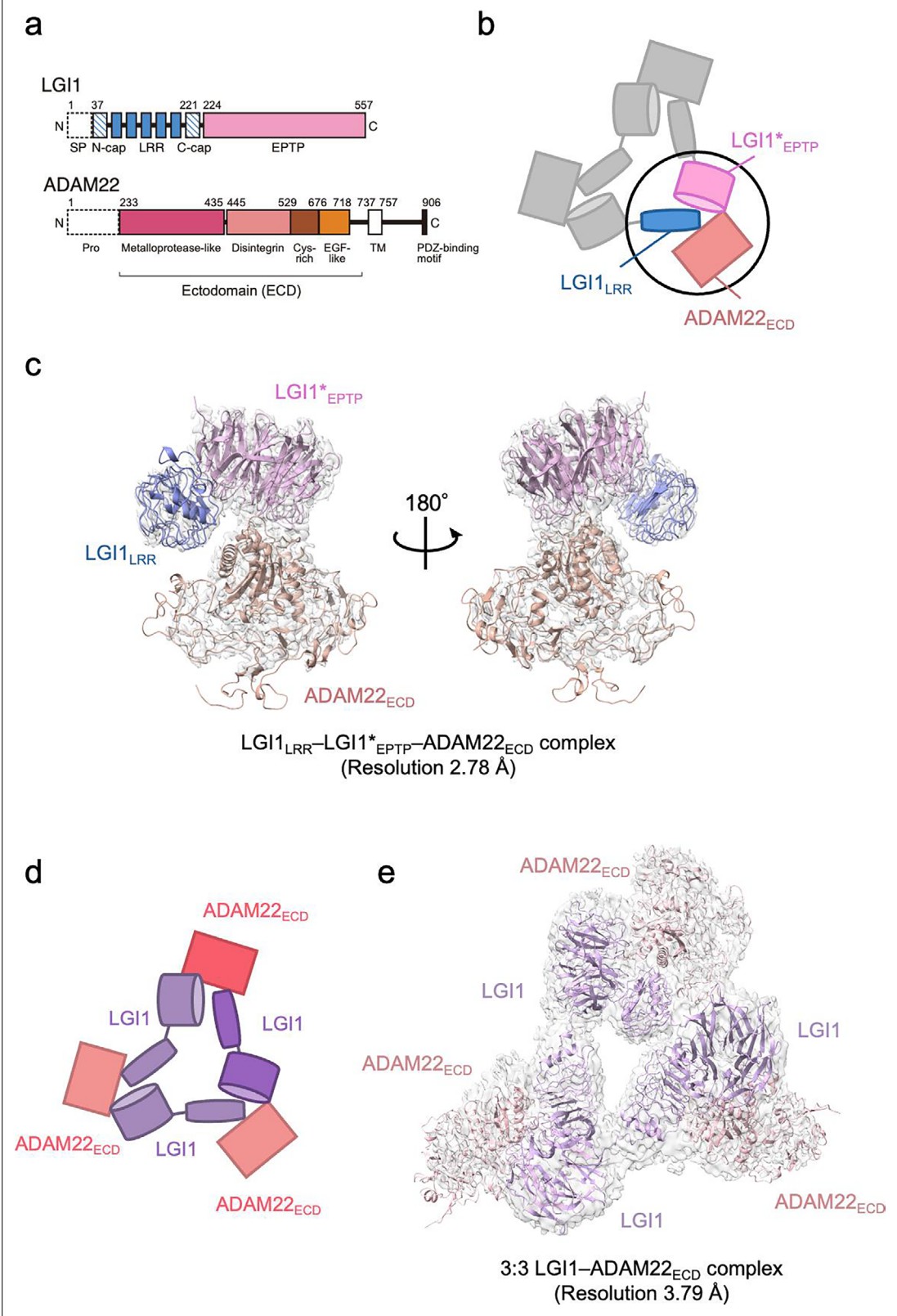

**Figure 1.** Structure of the LGI1–ADAM22_ECD complex. (**a**) Domain organizations of LGI1 and ADAM22. LGI1 consists of the leucine-rich repeat (LRR) (blue) and epitempin-repeat (EPTP) (pink) domains. The N-terminal secretion signal peptide (SP, enclosed by dotted lines) is removed in the secreted LGI1. The shaded blue boxes represent the N- and C-terminal caps, whereas the filled blue boxes represent the LRRs. The premature form of ADAM22 contains the N-terminal prosequence (enclosed by dotted lines). The mature ADAM22 consists of the metalloprotease-like (magenta), disintegrin

*Figure 1 continued on next page*

*Figure 1 continued*

(salmon pink), cysteine-rich (brown), EGF-like (orange), transmembrane (TM; white), and cytoplasmic domains. The major ADAM22 isoform has a PDZ-binding motif in the C-terminal of the cytoplasmic domain. (**b**) Schematic diagram of the LGI1$_{LRR}$–LGI1*$_{EPTP}$–ADAM22$_{ECD}$ complex. The black circle indicates the location of this complex within the 3:3 LGI1–ADAM22 complex. * indicates a distinct molecule. (**c**) Cryo-electron microscopy (cryo-EM) map and structure of the LGI1$_{LRR}$–LGI1*$_{EPTP}$–ADAM22$_{ECD}$ complex at 2.78 Å resolution. (**d**) Schematic diagram of the 3:3 LGI1–ADAM22$_{ECD}$ complex. (**e**) Cryo-EM map and structure of the 3:3 LGI1–ADAM22$_{ECD}$ complex at 3.79 Å resolution.

The online version of this article includes the following source data and figure supplement(s) for figure 1:

**Figure supplement 1.** Purification of the 3:3 LGI1–ADAM22$_{ECD}$ complex.

**Figure supplement 1—source data 1.** Original files for SDS-PAGE analysis displayed in *Figure 1—figure supplement 1b* (P1080457.RW2) and *Figure 1—figure supplement 1d* (P1080458.RW2).

**Figure supplement 1—source data 2.** PDF file containing uncropped images of SDS-PAGE gels for *Figure 1—figure supplement 1b and d*, indicating the relevant bands.

**Figure supplement 2.** Two-dimensional (2D) class averages of the particles from blob picker in CryoSPARC.

**Figure supplement 3.** Two-dimensional (2D) class averages of the particles from template picker in CryoSPARC.

**Figure supplement 4.** Cryo-electron microscopy (cryo-EM) data and processing for the LGI1$_{LRR}$–LGI1*$_{EPTP}$–ADAM22$_{ECD}$ and 3:3 LGI1–ADAM22$_{ECD}$ complexes.

*Sumino et al., 2024*) to directly visualize the molecular dynamics of the LGI1–ADAM22$_{ECD}$ assembly and characterize its structural property. The cryo-EM and HS-AFM results suggest that the 3:3 LGI1–ADAM22$_{ECD}$ assembly does not form a rigid threefold symmetric structure but rather a flexible triangular structure accompanying relative motions between each protomer.

## Results

### Cryo-EM single-particle analysis of the LGI1–ADAM22$_{ECD}$ complex

As we reported previously, the molar mass of the complex between the full-length LGI1 and ADAM22$_{ECD}$ determined by SEC-MALS at 150 mM NaCl was 356 kDa, corresponding to the 3:3 hexameric assembly of LGI1–ADAM22$_{ECD}$ in solution (*Yamagata et al., 2018*). This is consistent with our initial cryo-EM analysis, where 5% of the reference-free 2D class averaged images showed particles with pseudo-C3 symmetry suggestive of the 3:3 assembly. Since the 3:3 complex class was clearly seen and appeared structurally stable, we tried to determine the 3D structure of the 3:3 LGI1–ADAM22$_{ECD}$ complex by single-particle analysis but failed due to insufficient numbers of particles of the 3:3 complex. On this basis, we decided to perform a structural analysis of the complex stabilized by chemical cross-linking. We co-expressed His$_6$-tagged LGI1 and non-tagged ADAM22$_{ECD}$ in Expi293F cells and purified the complex using Ni-NTA affinity chromatography and gel filtration chromatography at 50 mM NaCl (*Figure 1—figure supplement 1a and b*, *Figure 1—figure supplement 1—source data 1*, *Figure 1—figure supplement 1—source data 2*). Then, fractions containing the LGI1–ADAM22$_{ECD}$ complex were collected and treated with glutaraldehyde to chemically cross-link the 3:3 complex. The sample was purified again by gel filtration chromatography. The chromatogram showed a peak likely corresponding to the 3:3 LGI1–ADAM22$_{ECD}$ complex (*Figure 1—figure supplement 1c and d*, *Figure 1—figure supplement 1—source data 1*, *Figure 1—figure supplement 1—source data 2*). One of the peak fractions was collected and subjected to single-particle analysis by cryo-EM. Among 2,061,420 particles picked from 7625 movies without templates, 1,403,037 particle images were extracted with a box size of 544 pixels (0.752 Å/pixel) and downsampled to a size of 136 pixels (3.008 Å/pixel) by Fourier cropping. The downsampled images were subjected to reference-free 2D classification, which generated images of triangle-shaped particles considered to be the 3:3 LGI1–ADAM22$_{ECD}$ complex (*Figure 1—figure supplement 2*). Other images looked like two particles stacked on top of each other or single particles clearly smaller than the 3:3 complex (*Figure 1—figure supplement 2*). The stacked particle images might represent the 2:2 complex viewed along the long axis or the 3:3 complex viewed from the side, which were difficult to distinguish. Eight classes with 176,443 particles of the putative 3:3 LGI1-ADAM22$_{ECD}$ complex were used as templates for picking particles again. 2,530,790 particle images were extracted, downsampled, and subjected to the second run of reference-free 2D classification, resulting in a set of images similar to that generated in the first run of 2D classification (*Figure 1—figure supplement 3*). Eighty 2D classes with 2,006,398 particles

**Table 1.** Data collection/processing and refinement statistics of cryo-electron microscopy (cryo-EM) single-particle analysis.

| | LGI1$_{LRR}$–LGI1*$_{EPTP}$–ADAM22$_{ECD}$ complex | 3:3 LGI1–ADAM22$_{ECD}$ complex |
|---|---|---|
| **Data collection and processing** | | |
| Magnification | 60,000 | 60,000 |
| Voltage (kV) | 300 | 300 |
| Dose rate (e⁻/pixel/s) | 10.6743 | 10.6743 |
| Defocus range (μm) | –0.8 to –2.2 | –0.8 to –2.2 |
| Pixel size (Å) | 0.752 | 0.752 |
| Symmetry imposed | C1 | C1 |
| Initial particle images (no.) | 2,061,420 | 2,061,420 |
| Final particle images (no.) | 557,450 | 120,728 |
| Map resolution (Å) | 2.78 | 3.78 |
| FSC threshold | 0.143 | 0.143 |
| | | |
| **Refinement** | | |
| Model resolution (Å) | | |
| FSC 0.143, unmasked/masked | 2.76/2.73 | 3.91/3.81 |
| Model composition | | |
| Non-hydrogen atoms | 7846 | 23,526 |
| Protein residues | 993 | 2979 |
| Ligands | 4 | 0 |
| B factors (Å²) | | |
| Protein (Å²) | 58.78 | 43.80 |
| Ligands (Å²) | 51.14 | |
| RMS deviation | | |
| Bond length (Å) | 0.002 | 0.012 |
| Bond angle (°) | 0.474 | 1.153 |
| MolProbity score | 1.45 | 2.28 |
| Clash score | 6.47 | 24.79 |
| Rotamer outliers (%) | 0.00 | 0.60 |
| Ramachandran plot | | |
| Favored (%) | 97.56 | 94.16 |
| Allowed (%) | 2.44 | 5.84 |
| Disallowed (%) | 0.00 | 0.00 |

were selected for ab initio 3D reconstruction. Six classes of ab initio 3D models were generated and then refined by heterogeneous refinement. The resultant 3D map in one class corresponded to a complex comprising LGI1$_{LRR}$, LGI1*$_{EPTP}$, and ADAM22$_{ECD}$ (* indicates a distinct molecule hereafter). Nonuniform refinement using the particle images with the original pixel size yielded a density map at a nominal resolution of 2.78 Å (*Figure 1b and c*, *Figure 1—figure supplement 4*, and *Table 1*). This map corresponds to a part of the 3:3 complex, where two-thirds of the complex may adopt various conformations. The 3D map of one other class corresponded to the 3:3 LGI1–ADAM22$_{ECD}$ complex. Nonuniform refinement using the particle images with the original pixel size yielded a density map at

a nominal resolution of 3.79 Å (*Figure 1d and e*, *Figure 1—figure supplement 4*, and *Table 1*). Using these two maps, we constructed atomic models of the LGI1$_{LRR}$–LGI1*$_{EPTP}$–ADAM22$_{ECD}$ complex and the 3:3 LGI1-ADAM22$_{ECD}$ complex (*Table 1*). The resolutions of the present cryo-EM analysis (2.78 Å and 3.79 Å) are better than that of the crystal structure of the 2:2 complex (7.125 Å).

## Interaction of the LGI1$_{EPTP}$–ADAM22$_{ECD}$ complex

In the cryo-EM structure of the LGI1$_{LRR}$–LGI1*$_{EPTP}$–ADAM22$_{ECD}$ complex, the LGI1$_{EPTP}$–ADAM22$_{ECD}$ structure is essentially identical to that determined by X-ray crystallography (*Figure 2a*). At the interface, Trp398, Tyr408, and Tyr409 of ADAM22 are stacked in a layer and project into the hydrophobic inner rim of the central channel of LGI1$_{EPTP}$, which consists of Leu237, Phe256, Val284, Leu302, Tyr433, Met477, and Phe541 of LGI1 (*Figure 2a*). In addition, several hydrogen bonds are formed between LGI1 and ADAM22: Arg330 and Lys353 of LGI1 form hydrogen bonds with Asp405 and Glu359 of ADAM22, respectively, whereas Arg378 of LGI1 forms hydrogen bonds with Ser340 and Thr406 of ADAM22 (*Figure 2b and c*, *Figure 2—figure supplement 1*). The hydrogen bond between Asp431 of LGI1 and Lys362 of ADAM22 was also found in the cryo-EM structure (*Figure 2—figure supplement 1*). These hydrogen bonds differ slightly from those in the previous crystal structure: in the previous crystal structure, Lys331 of LGI1 formed a hydrogen bond with the main-chain carbonyl of Asp405 of ADAM22. On the other hand, in the present cryo-EM structure, Lys331 of LGI1 is reoriented and does not form a hydrogen bond (*Figure 2b*). Arg378 of LGI1 hydrogen bonds with Ser340 and Glu359 in the crystal structure, while it does with Ser340 and Thr406 in the cryo-EM structure (*Figure 2c*). In the crystal structure, Asp431 of LGI1 faces outward and does not form a hydrogen bond with Lys362 of ADAM22 (*Figure 2—figure supplement 1*). These differences in hydrogen bonding may reflect the varying contribution of each interacting residue to the affinity between LGI1$_{EPTP}$ and ADAM22$_{ECD}$. As shown in our previous pull-down experiments, mutations of the hydrophobic residues at the interface of ADAM22 (i.e. Trp398, Tyr408, and Tyr409) almost or completely abolished binding to LGI1, whereas the E359A or D405A mutation of ADAM22 decreased but did not abolish binding to LGI1 (*Yamagata et al., 2018*). The hydrogen bonds involved in the LGI1$_{EPTP}$–ADAM22$_{ECD}$ interaction might be so weak that the orientation of the hydrogen bonding residues could alter. In the cryo-EM structure, we also found hydrogen bonds between Ser282 of LGI1 and the main-chain carbonyl of Thr397 of ADAM22, between the main-chain carbonyl of Trp376 of LGI1 and Gln334 of ADAM22, and between Lys353 of LGI1 and the main-chain carbonyl of Phe335 of ADAM22 (*Figure 2—figure supplement 1*). These three hydrogen bonds were also formed in the crystal structure.

## Intermolecular interactions between LGI1$_{LRR}$ and LGI1*$_{EPTP}$

The present cryo-EM map of the LGI1$_{LRR}$–LGI1*$_{EPTP}$–ADAM22$_{ECD}$ complex provides a more detailed view of the interaction between LGI1$_{LRR}$ and LGI1*$_{EPTP}$ (*Figure 3a*) than the previous crystal structure of the 2:2 complex at a moderate resolution of 7.125 Å. Specifically, Glu123 and Arg76 of LGI1$_{LRR}$ form hydrogen bonds with Arg474 and Glu516 of LGI1*$_{EPTP}$, respectively. The hydrogen bond between Glu123 of LGI1$_{LRR}$ and Arg474 of LGI*$_{EPTP}$ is reinforced by stacking with Phe121 of LGI1$_{LRR}$ (*Figure 3a*). Additionally, Asn52 and Ser73 of LGI1$_{LRR}$ form hydrogen bonds with the main-chain carbonyls of Tyr496 and Asp495 of LGI1*$_{EPTP}$, respectively. Leu54, Val75, Leu97, and Phe121 of LGI1$_{LRR}$ also form extensive hydrophobic interactions (*Figure 3a*). The R474Q mutation in LGI1$_{EPTP}$ is a missense ADEAF mutation in LGI1, known to cause epileptic symptoms by inhibiting the assembly of the LGI1–ADAM22 higher-order complex (*Dazzo et al., 2016*; *Kawamata et al., 2010*; *Yamagata et al., 2018*). The present higher-resolution structure demonstrates that Arg474 of LGI1*$_{EPTP}$ forms a hydrogen bond with Glu123 of LGI1$_{LRR}$, which was ambiguous in the previous moderate-resolution crystal structure. Regarding interactions for the higher-order assembly, the previous crystal structure suggested a possible interaction between His116 of LGI1$_{LRR}$ and Glu446 of ADAM22 (*Yamagata et al., 2018*). However, in the present cryo-EM structure, the distance between His116 and Glu446 is 6.6 Å (*Figure 3—figure supplement 1*), indicating that they do not interact with each other. No tight interaction was found between LGI1$_{LRR}$ and ADAM22$_{ECD}$ in the cryo-EM structure of the LGI1$_{LRR}$–LGI1*$_{EPTP}$–ADAM22$_{ECD}$ complex.

## Cryo-EM structure of the 3:3 LGI1–ADAM22$_{ECD}$ complex

The cryo-EM structure of the LGI1$_{LRR}$–LGI1*$_{EPTP}$–ADAM22$_{ECD}$ complex could fit a density map of the 3:3 LGI1–ADAM22$_{ECD}$ complex well. Three LGI1$_{LRR}$–LGI1*$_{EPTP}$–ADAM22$_{ECD}$ complex structures were put

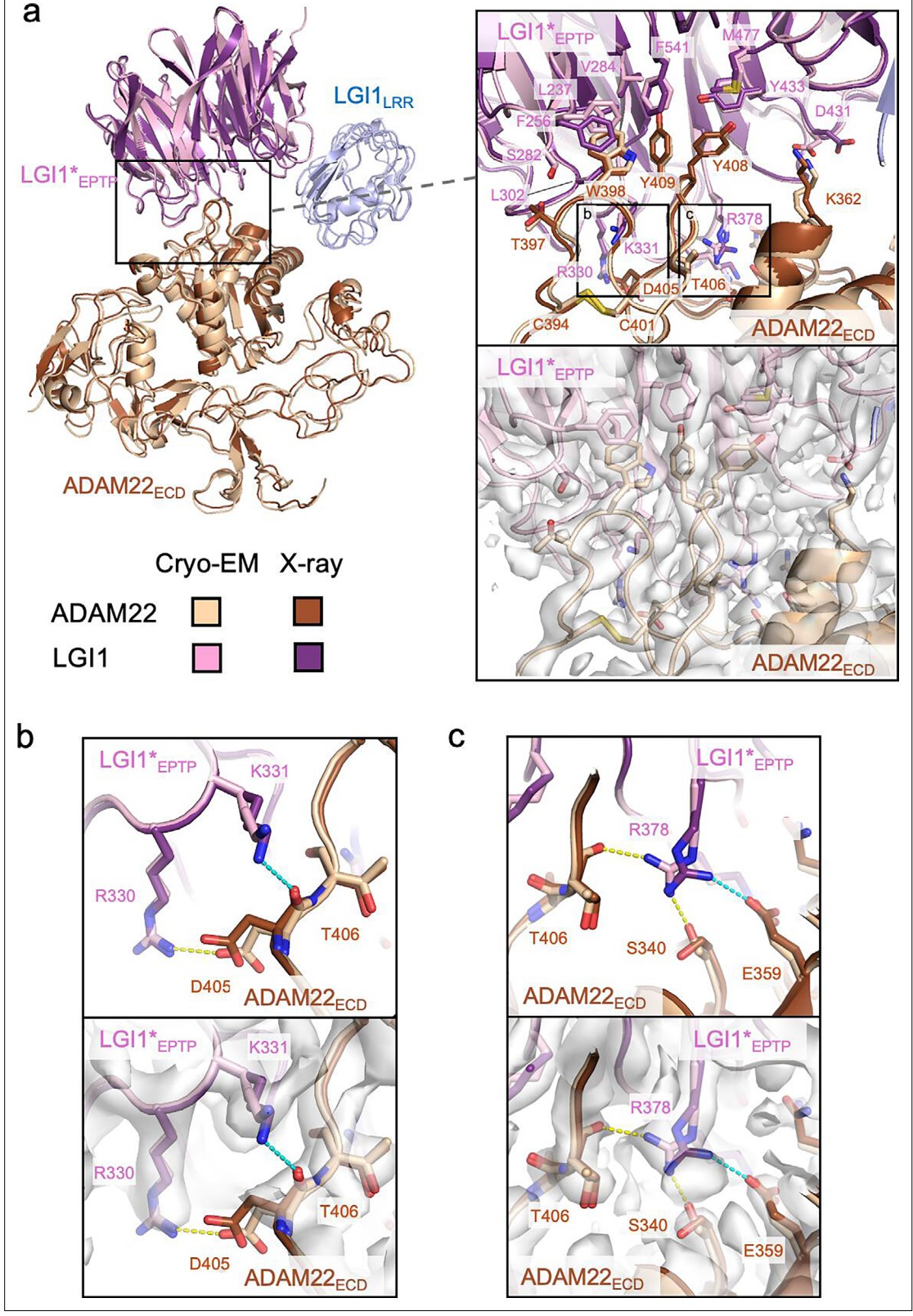

**Figure 2.** Interactions between LGI1$_{EPTP}$ and ADAM22$_{ECD}$. (**a**) Overall view of the cryo-electron microscopy (cryo-EM) structure of the LGI1$_{LRR}$–LGI1*$_{EPTP}$–ADAM22$_{ECD}$ complex and a magnified view of the interface between LGI1*$_{EPTP}$ and ADAM22$_{ECD}$. The density map of the interface is shown as white surfaces. The previously reported X-ray structure of the LGI1$_{EPTP}$–ADAM22$_{ECD}$ complex (PDB 5Y2Z) is superposed with distinct colors. Two boxes in the magnified view indicate the locations of the views shown in (**b**, left box) and (**c**, right box). (**b**) Close-up view of the interaction between Arg330

*Figure 2 continued on next page*

Figure 2 continued

of LGI1*$_{EPTP}$ and Asp405 of ADAM22$_{ECD}$. The structure (top) and corresponding map (bottom) are shown. Yellow and cyan dashed lines indicate a hydrogen bond observed in the cryo-EM and X-ray structures, respectively. (**c**) Close-up view of the interaction around Arg378 of LGI1. The structure (top) and corresponding map (bottom) are shown. Yellow and cyan dashed lines indicate hydrogen bonds observed in the cryo-EM and X-ray structures, respectively.

The online version of this article includes the following figure supplement(s) for figure 2:

**Figure supplement 1.** Comparison of the LGI1$_{EPTP}$–ADAM22$_{ECD}$ interaction between the cryo-electron microscopy (cryo-EM) and crystal structures.

into the 3.79-Å-resolution map of the 3:3 complex. The density of LGI1 and the metalloprotease-like domain of ADAM22$_{ECD}$ appears relatively strong. The density of one interface between LGI1*$_{EPTP}$ and ADAM22$_{ECD}$ is especially resolved well (*Figure 3—figure supplement 2*; see the interface between chains A and B). Although the density of amino acid side chains in LGI1$_{LRR}$ is not well separated, the main-chain structure of LGI1$_{LRR}$ could be fitted to the density map (*Figure 3—figure supplement 2*; see the density of LGI1$_{LRR}$ in chain F). On the other hand, the density of the disintegrin, cysteine-rich, and EGF-like domains after Pro445 of ADAM22 is relatively weak in all three molecules. Correspondingly, in the local resolution map of the 3:3 LGI1–ADAM22$_{ECD}$ complex, the resolution of these ADAM22 domains after Pro445 is low (*Figure 3—figure supplement 3*). Like the crystal structure of the 2:2 LGI1–ADAM22$_{ECD}$ complex, the LRR and EPTP domains of LGI1 are linked by a two-residue linker (Ile222–Ile223) in an extended conformation. LGI1$_{EPTP}$ interacts with the metalloprotease-like domain of ADAM22 to form the LGI1$_{EPTP}$–ADAM22$_{ECD}$ complex, while the LRR domain of one LGI1 molecule interacts with the EPTP domain of the neighboring LGI1, thereby bridging three distant ADAM22 molecules in the 3:3 complex (*Figure 1d and e*). The C-termini of the three ADAM22 molecules point in opposite directions. When the three assembled LGI1–ADAM22$_{ECD}$ complexes were superposed with the LGI1$_{EPTP}$–ADAM22$_{ECD}$ structure as the reference, all three LGI1 LRR domains were oriented differently (*Figure 3b and c*). Although the triangular shape observed in the 2D class-averaged image suggested (pseudo-)$C3$ symmetry of the 3:3 complex, the determined structure of the 3:3 complex was not symmetric. Actually, the $C3$ symmetry-restrained 3:3 model that we previously calculated based on the SEC-SAXS analysis using the program SASREF (*Petoukhov and Svergun, 2005*; *Yamagata et al., 2018*) could not be fitted with the present cryo-EM structure (*Figure 3—figure supplement 4a and b*). This discrepancy arises from the difference in the orientation of LGI1$_{LRR}$ relative to the LGI1*$_{EPTP}$–ADAM22 complex between the previous model and the present cryo-EM structure (indicated by the arrowhead in *Figure 3—figure supplement 4b*). The structure of the 3:3 LGI1–ADAM22 complex was also predicted by AlphaFold3 (*Abramson et al., 2024*), which suggested a $C3$ symmetric assembly (*Figure 3—figure supplement 4a*). Intriguingly, the orientation of LGI1$_{LRR}$ relative to LGI1*$_{EPTP}$–ADAM22 was similar to that observed in the present cryo-EM structure (indicated by the arrowhead in *Figure 3—figure supplement 4c*), despite relatively low prediction accuracy of the assembly (ipTM = 0.47, pTM = 0.52; pLDDT color outputs and PAE plot are shown in *Figure 3—figure supplement 4d and e*). On the other hand, the overall trimeric configuration is different between the $C3$ symmetric AlphaFold3 model and the nonsymmetric cryo-EM structure (*Figure 3—figure supplement 4c*).

We then analyzed the interdomain motion of the three LGI1 domains (assigned to chains B, D, and F in the deposited PDB file; *Figure 3b*) in the 3:3 LGI1–ADAM22$_{ECD}$ complex by the DynDom server (*Veevers and Hayward, 2019*; *Figure 3—figure supplement 5*, and *Table 2*). Analysis of the motion between the three pairs of chains suggested that the static domain corresponds to the EPTP domain, while the mobile domain corresponds to the LRR domain, predicting a hinge region for interdomain bending. The LRR domain of chain D was rotated 69.8° around the hinge axis compared to chain B, and the LRR domain of chain F was rotated 69.0° around the hinge axis compared to chain B. The LRR domains of chain D and chain F were also rotated 70.0° around the hinge axis relative to each other, suggesting that each LRR domain rotates approximately 70° about the hinge axis in the 3:3 LGI1–ADAM22$_{ECD}$ complex (*Table 2*). Furthermore, the motion of each LRR domain varied widely: the movement of the LRR domain of chain D with respect to chain B follows a standard closure motion of 14%, whereas the movement of the LRR domain of chain F with respect to chain D follows a standard closure motion of 99% (*Table 2*). This closure motion appears to locate the LRR domain of chain F in close proximity to that of chain D to make the triangular assembly slightly more compact, which might stabilize the nonsymmetric trimeric configuration observed in the cryo-EM structure.

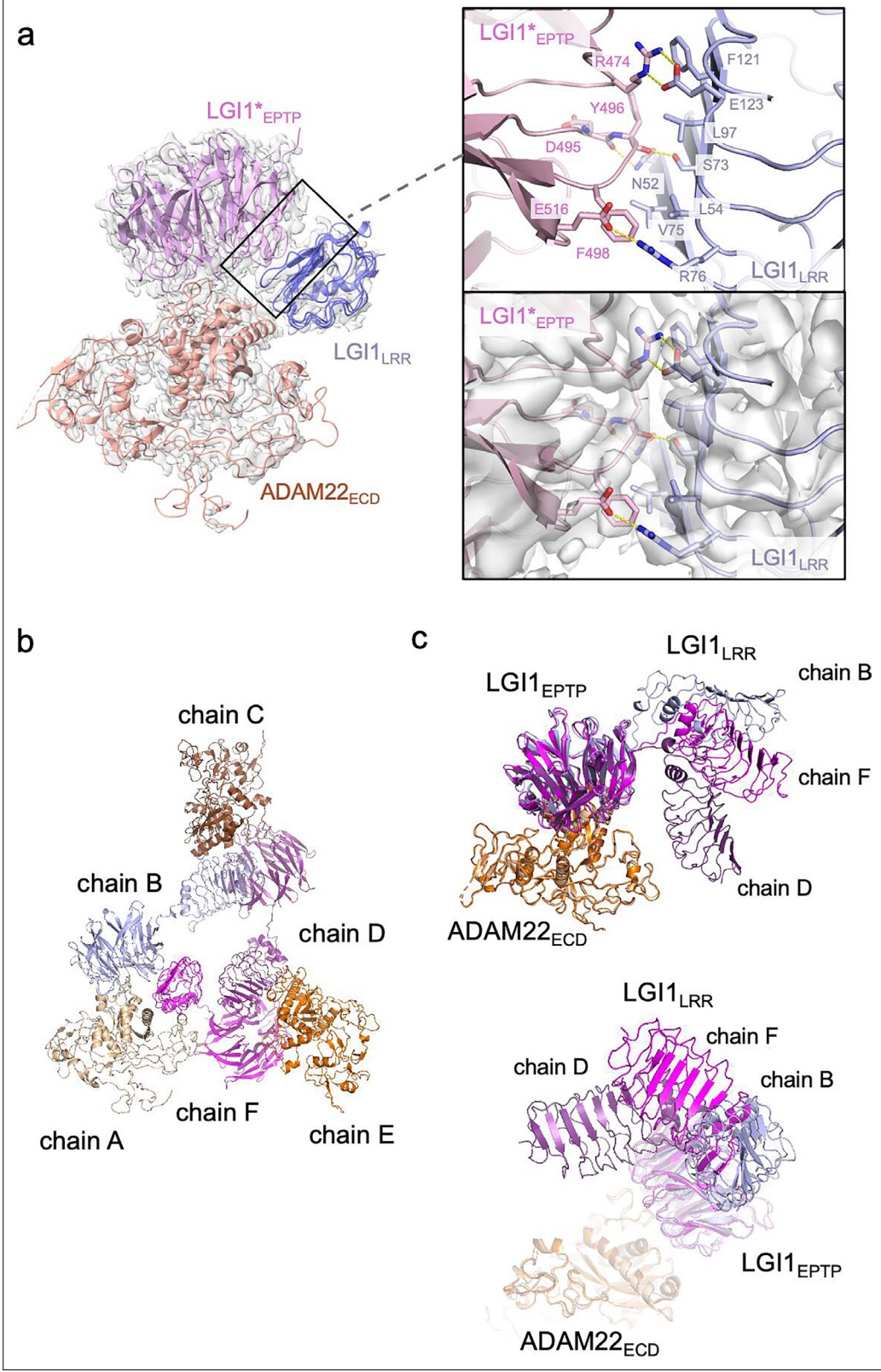

**Figure 3.** Structure and interaction of the higher-order LGI1–ADAM22$_{ECD}$ complex. (**a**) Overall view of the cryo-electron microscopy (cryo-EM) structure of the LGI1$_{LRR}$–LGI1*$_{EPTP}$–ADAM22$_{ECD}$ complex and a magnified view of the interface between LGI1$_{LRR}$ and LGI1*$_{EPTP}$. The density map of the interface is shown as white surfaces. (**b**) Chain IDs of the individual LGI1 or ADAM22$_{ECD}$ molecules in the 3:3 LGI1–ADAM22$_{ECD}$ complex, assigned in this study.

*Figure 3 continued on next page*

*Figure 3 continued*

(**c**) Superposition of the three LGI1–ADAM22$_{ECD}$ complexes in the cryo-EM structure of the 3:3 LGI1-ADAM22$_{ECD}$ complex, using LGI1$_{EPTP}$–ADAM22$_{ECD}$ as the reference.

The online version of this article includes the following figure supplement(s) for figure 3:

**Figure supplement 1.** LGI1$_{LRR}$ and ADAM22$_{ECD}$ in the LGI1–ADAM22$_{ECD}$ assembly.

**Figure supplement 2.** Cryo-electron microscopy (cryo-EM) density map of the 3:3 LGI1–ADAM22$_{ECD}$ complex.

**Figure supplement 3.** Local resolution of the cryo-electron microscopy (cryo-EM) structure of the 3:3 LGI1–ADAM22$_{ECD}$ complex.

**Figure supplement 4.** Comparison of the 3:3 LGI1–ADAM22$_{ECD}$ assembly determined by cryo-electron microscopy (cryo-EM) with that calculated based on size-exclusion chromatography-small-angle X-ray scattering (SEC-SAXS) analysis and that predicted by AlphaFold3.

**Figure supplement 5.** Conformational difference of the three LGI1 molecules in the 3:3 LGI1–ADAM22$_{ECD}$ assembly.

## Dynamics of the LGI1–ADAM22 higher-order complex observed by HS-AFM

To directly visualize the molecular dynamics of the LGI1–ADAM22$_{ECD}$ complex and characterize its structural properties in solution, we performed HS-AFM (*Ando et al., 2001*; *Sumino et al., 2024*). HS-AFM images of gel filtration chromatography fractions containing the 3:3 LGI1–ADAM22$_{ECD}$ complex (not chemically cross-linked with glutaraldehyde) predominantly revealed triangular-shaped molecules, which appeared to exist stably with no drastic structural changes (*Figure 4a–c* and *Figure 4—video 1*). A comparison with the simulated AFM image suggests that the protrusion on the exterior of the triangle is likely ADAM22 (*Figure 4b*). This site frequently appeared to disso-ciate during HS-AFM scanning (*Figure 4c* and *Figure 4—video 1*), indicating that the interaction between LGI1 and ADAM22 is weaker than the interactions among LGI1 molecules within the 3:3

**Table 2.** Domain motion analysis of LGI1 in the 3:3 LGI1–ADAM22$_{ECD}$ complex by the DynDom server.

| Chain IDs | DynDom parameters | LGI1 |
|---|---|---|
| | Fixed domain | Residues 225–549 (RMSD 1.79 Å) |
| | Moving domain | Residues 43–224 (RMSD 1.16 Å) |
| | Rotation angle (°) | 69.8 |
| | Translation (Å) | −1.8 |
| | Closure (%) | 13.8 |
| B vs D | Bending residues | 215–225 |
| | Fixed domain | Residues 223–549 (RMSD 1.01 Å) |
| | Moving domain | Residues 43–222 (RMSD 1.19 Å) |
| | Rotation angle (°) | 69.0 |
| | Translation (Å) | −0.8 |
| | Closure (%) | 52.8 |
| B vs F | Bending residues | 215–223 |
| | Fixed domain | Residues 224–549 (RMSD 1.82 Å) |
| | Moving domain | Residues 43-223 (RMSD 1.24 Å) |
| | Rotation angle (°) | 70.0 |
| | Translation (Å) | −1.7 |
| | Closure (%) | 99.0 |
| D vs F | Bending residues | 218–225 |

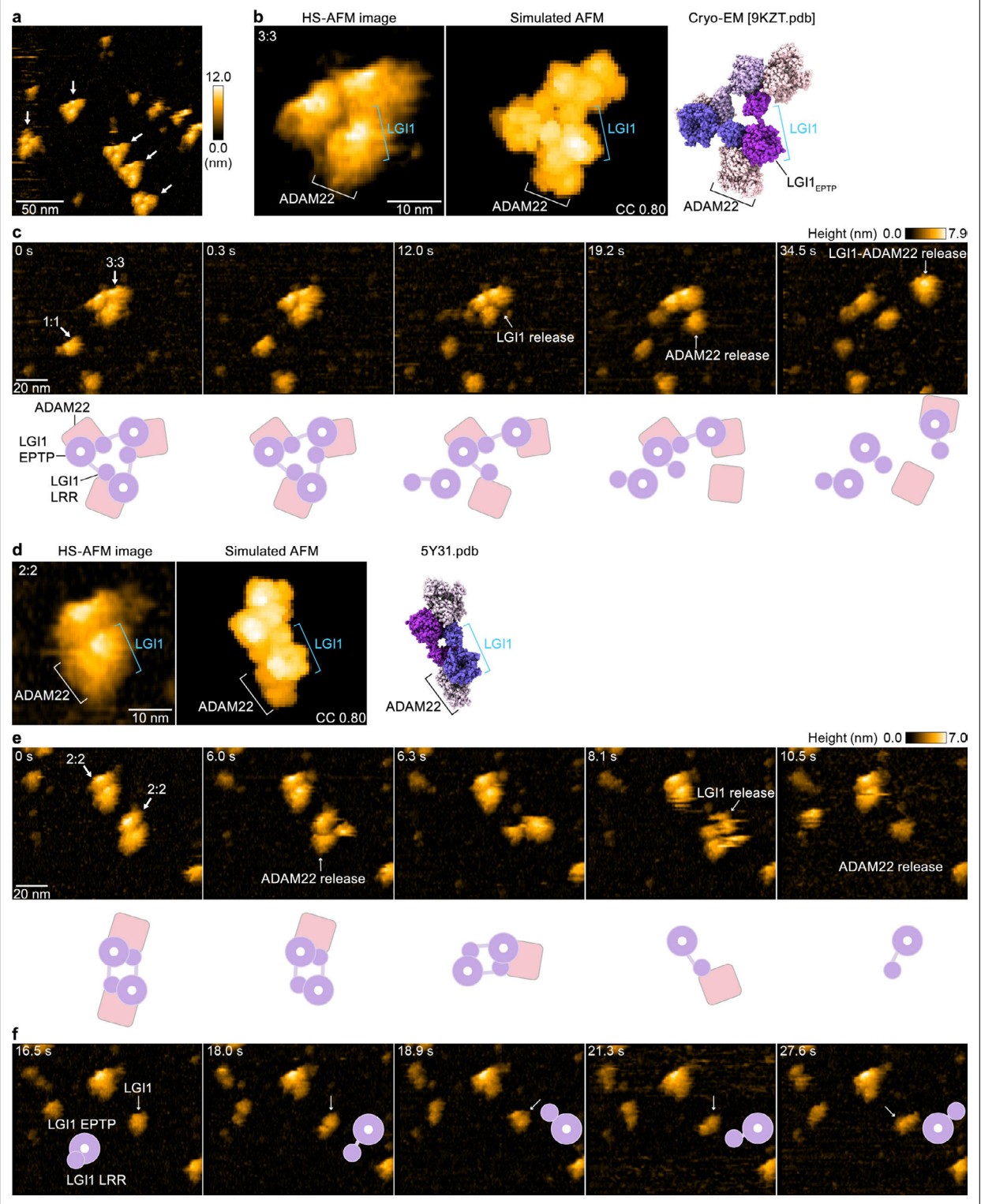

**Figure 4.** High-speed atomic force microscopy (HS-AFM) observations of LGI1–ADAM22$_{ECD}$ complexes. (**a**) A representative HS-AFM image of the 3:3 LGI1–ADAM22$_{ECD}$ complex. The color bars on the right indicate height in nanometers. White arrows indicate the 3:3 complex. The frame rate was 1.0 frames/s. (**b, d**) Magnified HS-AFM images of the 3:3 (left in **b**) and 2:2 (left in **d**) LGI1–ADAM22$_{ECD}$ complexes. The simulated AFM images (middle) were derived from fitting to the experimental HS-AFM image (left). The well-fitting simulated AFM images and the coefficient of correlation (CC) are indicated. The cryo-electron microscopy (cryo-EM) structure of the 3:3 complex (right in **b**) and the X-ray structure of the 2:2 complex (right in **d**) are shown in the same orientation as the simulated AFM images. (**c, e, f**) Sequential HS-AFM images of the 3:3 (c; see also ***Figure 4—video 1***) and 2:2

*Figure 4 continued on next page*

*Figure 4 continued*

(**e**, **f**; see also *Figure 4—video 2*) LGI1–ADAM22$_{ECD}$ complexes. A schematic illustration of the interpretation of HS-AFM images is shown at the bottom. Imaging parameters: scanning area = 120 × 96 nm$^2$ (240×192 pixels); frame rate = 3.3 frames/s. HS-AFM experiments were repeated independently at least three times with consistent results.

The online version of this article includes the following video(s) for figure 4:

**Figure 4—video 1.** High-speed atomic force microscopy (HS-AFM) videos of three representative 3:3 LGI1–ADAM22$_{ECD}$ complexes on the AP-mica, related to *Figure 4c*.

https://elifesciences.org/articles/105918/figures#fig4video1

**Figure 4—video 2.** High-speed atomic force microscopy (HS-AFM) videos of a representative 2:2 LGI1–ADAM22$_{ECD}$ complex on the AP-mica, related to *Figure 4e*.

https://elifesciences.org/articles/105918/figures#fig4video2

LGI1–ADAM22$_{ECD}$ assembly. Together with the cryo-EM structure, this also indicates that the trimerization can be entirely organized by LGI1, suggesting the possibility that LGI1 could trimerize on its own, although this possibility could not be tested due to the difficulty in the expression of the full-length LGI1 alone for biophysical analysis in our hands. On the other hand, considering the dynamic property of the 3:3 complex and spatial alignment of LGI1$_{LRR}$ and ADAM22$_{ECD}$, we cannot exclude the possibility that ADAM22 could act as a platform to facilitate the intermolecular interaction between LGI1$_{LRR}$ and LGI1*$_{EPTP}$ for the trimerization of LGI1.

In addition to the 3:3 complex, HS-AFM also captured the dynamics of the 2:2 LGI1–ADAM22$_{ECD}$ complex (*Figure 4d and e* and *Figure 4—video 2*). A comparison of the experimental HS-AFM image with the simulated AFM image based on the crystal structure of the 2:2 complex indicates that the outer site corresponds to ADAM22, suggesting that LGI1 and ADAM22 are facing each other. The 2:2 complex also exhibited fragility during HS-AFM imaging, similar to the 3:3 complex (*Figure 4e* and *Figure 4—video 2*). In individual LGI1 molecules dissociated from the complex, the LRR domain moved freely relative to the EPTP domain (*Figure 4f* and after 10.2 s in *Figure 4—video 2*), likely reflecting the conformational difference in the relative orientation between LGI1$_{LRR}$ and LGI1$_{EPTP}$ observed within the cryo-EM structure of the 3:3 complex. Thus, single-molecule observations using HS-AFM demonstrated that the 3:3 LGI1–ADAM22$_{ECD}$ complex is present in solution, that the binding between ADAM22 and LGI1 is relatively weak within the 3:3 LGI1–ADAM22$_{ECD}$ assembly, and that the LRR domain of LGI1 exhibits flexibility.

## Discussion

In this study, the ligand–receptor complex between LGI1, a secreted protein of neurons, and its receptor protein, ADAM22, was investigated by cryo-EM and HS-AFM. By chemically cross-linking with glutaraldehyde, we successfully captured much larger numbers of particle images of the 3:3 LGI1–ADAM22$_{ECD}$ complex than in our initial, preliminary cryo-EM study (*Yamagata et al., 2018*). We could determine not only the overall 3D structure of the 3:3 LGI1–ADAM22$_{ECD}$ complex but also the 3D structure of the LGI1$_{LRR}$– LGI1*$_{EPTP}$–ADAM22$_{ECD}$ complex at a nominal resolution of 2.78 Å, revealing the interaction mode of the LGI1–ADAM22 higher-order complex at a higher resolution than before. HS-AFM successfully imaged the LGI1–ADAM22$_{ECD}$ higher-order complex and confirmed that LGI1–ADAM22$_{ECD}$ forms a 3:3 complex in solution.

The present 3.79-Å-resolution cryo-EM map of the 3:3 LGI1–ADAM22$_{ECD}$ complex was calculated from 120,708 particle images selected after

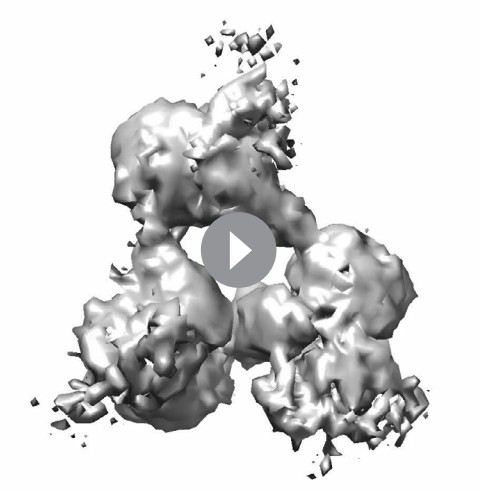

**Animation 1.** 3D flexible refinement (3D Flex) analysis of the 3:3 LGI1–ADAM22$_{ECD}$ complex. Two latent coordinates were used in the 3D Flex analysis. Two 3D Flex movies (*Animations 1 and 2*) are displayed at the same threshold.

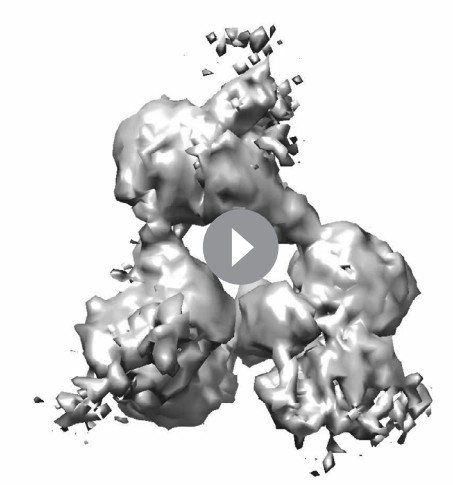

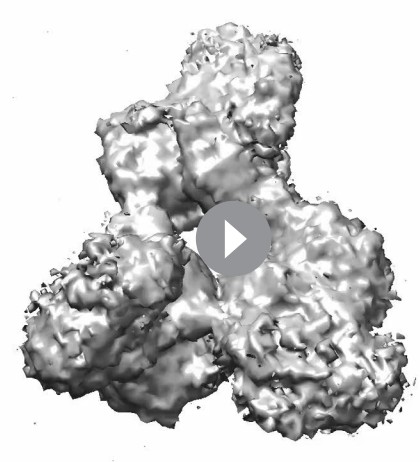

**Animation 2.** 3D flexible refinement (3D Flex) analysis of the 3:3 LGI1–ADAM22 complex. Two latent coordinates were used in the 3D Flex analysis. Two 3D Flex movies (*Animations 1 and 2*) are displayed at the same threshold.

**Animation 4.** 3D Variability Analysis (3D VA) of the 3:3 LGI1–ADAM22$_{ECD}$ complex. Three variability components are represented in the 3D VA. Three 3D VA movies (*Animations 3–5*) are displayed at the same threshold. Two movies (*Animations 3 and 4*) show twisting motion, whereas one movie (*Animation 5*) shows stretching motion.

two rounds of heterogeneous refinement. On the other hand, a few classes of other particle images display triangular shapes with missing parts, suggesting domain motions or conformational heterogeneity in the 3:3 complex (*Figure 1—figure supplement 4*). This raised the possibility that motion-based refinement might improve resolution in flexible regions. Therefore, we first performed 3D flexible refinement (3D Flex) using CryoSPARC (*Punjani and Fleet, 2023*; *Animation 1 and 2*). However, even with the 3D Flex refinement, the density of the disintegrin, cysteine-rich, and EGF-like domains in the ADAM22 molecule remained poorly resolved. Then, we next hypothesized that the 3:3 LGI1–ADAM22$_{ECD}$ complex undergoes discrete, non-rigid motions and performed 3D

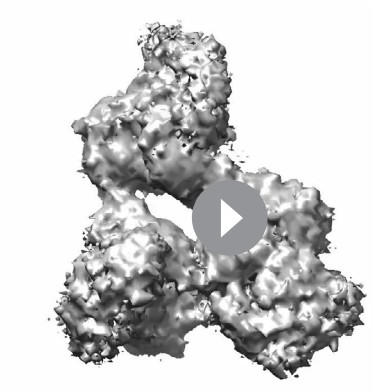

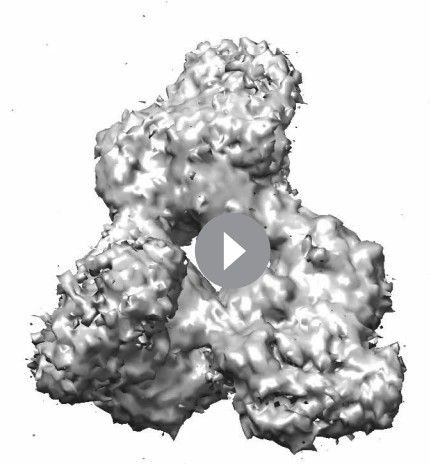

**Animation 3.** 3D Variability Analysis (3D VA) of the 3:3 LGI1–ADAM22$_{ECD}$ complex. Three variability components are represented in the 3D VA. Three 3D VA movies (*Animations 3–5*) are displayed at the same threshold. Two movies (*Animations 3 and 4*) show twisting motion, whereas one movie (*Animation 5*) shows stretching motion.

**Animation 5.** 3D Variability Analysis (3D VA) of the 3:3 LGI1–ADAM22$_{ECD}$ complex. Three variability components are represented in the 3D VA. Three 3D VA movies (*Animations 3–5*) are displayed at the same threshold. Two movies (*Animations 3 and 4*) show twisting motion, whereas this movie shows stretching motion.

Variability Analysis (3D VA) in CryoSPARC (*Punjani and Fleet, 2021*). Using three orthogonal principal modes, the 3D VA indicated two twisting motions and one stretching motion of the triangular-shaped 3:3 LGI1–ADAM22$_{ECD}$ complex (*Animation 3–5*). This analysis also visualized relatively large motions of the disintegrin, cysteine-rich, and EGF-like domains of ADAM22 (*Animation 3–5*), suggestive of intrinsic conformational flexibilities of these domains, likely resulting in lower local resolution in the periphery of the complex (*Figure 3—figure supplement 3*).

Previous studies have shown that LGI1 is enriched not only at the synapse but also at the axon initial segment and colocalized with ADAM22/23 and the voltage-gated potassium (Kv1) channels (*Hivert et al., 2019*; *Seagar et al., 2017*). The Kv1-associated cell-adhesion molecules, TAG-1 and Caspr2, likely mediate this colocalization by binding to ADAM22/23. LGI1 knockout mice show a reduced density of Kv1 channels, which is associated with increased neuronal excitability of hippocampal CA3 neurons, indicating that LGI1 regulates Kv1 channel function (*Seagar et al., 2017*). In addition, the LGI1 R474Q mutation has been found to interfere with the colocalization of ADAM22/23 and Kv1 channels (*Hivert et al., 2019*). Given that the LGI1 R474Q mutation inhibits the higher-order assembly of LGI1–ADAM22 with little impact on LGI1 secretion and binding to ADAM22, the higher-order complex of LGI1–ADAM22 likely regulates the axonal Kv1 channel function. The 3:3 LGI1–ADAM22 complex observed in vitro might facilitate efficient clustering of the axonal Kv1 channels to control their density and inhibit epilepsy. A recent study has shown that LGI3, a member of the LGI family, selectively co-assembles with Kv1 channels by using ADAM23 as the receptor in axons (*Miyazaki et al., 2024*). LGI3 is secreted by oligodendrocytes in the brain and enriched at juxtaparanodes of myelinated axons to form subclusters. LGI1 and LGI3, along with ADAM22 and ADAM23, belong to the same subfamily and share a similar domain organization. This suggests that the LGI3–ADAM23 complex, like the LGI1–ADAM22 complex, may form a higher-order assembly, which could be related to the clustering mechanism of Kv1 channels in axons. In this context, as discussed in *Miyazaki et al., 2024*, either or both of the 2:2 and 3:3 complexes might be formed in a *trans* fashion at the juxtaparanode of myelinated axons and bridge the axon and the innermost myelin membrane. Alternatively, the 3:3 complex formed in a *cis* fashion might positively regulate the clustering of the axonal Kv channels at the juxtaparanode, possibly in a similar manner at the axon initial segment. Finally, this study proposes that the LGI1–ADAM22 complex is an interesting therapeutic target for epilepsy and other neurological disorders. The 3:3 LGI1–ADAM22 complex structure revealed in this study could serve as a useful platform for structure-based drug design and facilitate the development of antiepileptic drugs.

# Materials and methods

**Key resources table**

| Reagent type (species) or resource | Designation | Source or reference | Identifiers | Additional information |
|---|---|---|---|---|
| Gene (*Homo sapiens*) | LGI1 | GenBank | NM_005097 | |
| Gene (*Homo sapiens*) | ADAM22 | GenBank | NM_021723 | |
| Cell line (*Homo sapiens*) | Expi293F | Thermo Fisher Scientific | Cat # A14527 | |
| Transfected construct (*Homo sapiens*) | Igκ-LGI1 (37–557; R470A) -His$_6$ (plasmid) | *Yamagata et al., 2018* (PMID:29670100) | | pEBMulti-Neo (backbone) |
| Transfected construct (*Homo sapiens*) | non-tagged ADAM22 (1–729) (plasmid) | *Yamagata et al., 2018* (PMID:29670100) | | pEBMulti-Neo (backbone) |
| Software, algorithm | SerialEM | *Mastronarde, 2005*; http://bio3d.colorado.edu/SerialEM/ | RRID:SCR_017293 | |
| Software, algorithm | yoneoLocr Version 1.0 | *Yonekura et al., 2021*; https://github.com/YonekuraLab/yoneoLoc | | |
| Software, algorithm | CryoSPARC Version 4.1.2 | *Punjani et al., 2017*; https://cryosparc.com | RRID:SCR_016501 | |

*Continued on next page*

*Continued*

| Reagent type (species) or resource | Designation | Source or reference | Identifiers | Additional information |
|---|---|---|---|---|
| Software, algorithm | Coot | *Emsley and Cotwan, 2004*; https://www2.mrc-lmb.cam.ac.uk/personal/pemsley/coot/ | RRID:SCR_014222 | |
| Software, algorithm | Phenix Version 1.19–4092 | *Adams et al., 2011*; https://phenix-online.org | RRID:SCR_014224 | |
| Software, algorithm | UCSF ChimeraX Version 1.15 | *Meng et al., 2023*; https://www.cgl.ucsf.edu/chimerax/ | RRID:SCR_015872 | |
| Software, algorithm | PyMOL Version 3.1.0 | Schrödinger, LLC; http://www.pymol.org | RRID:SCR_000305 | |
| Software, algorithm | Fiji (ImageJ) software | *Schindelin et al., 2012*; https://imagej.net/software/fiji/ | RRID:SCR_002285 | |
| Software, algorithm | BioAFMviewer | *Amyot and Flechsig, 2020*; https://www.bioafmviewer.com/index.php | | |
| Other | Expi293 Expression Medium | Thermo Fisher Scientific | Cat # A1435101 | Cell culture medium for Expi293F cells |
| Other | Quantifoil holey carbon grid | Quantifoil | R1.2/1.3, Cu, 300 mesh | Carbon holey grid for cryo-EM |

## Protein preparation

For preparation of the LGI1–ADAM22$_{ECD}$ complex, the C-terminally His$_6$-tagged LGI1 (R470A) was co-expressed with the non-tagged ADAM22$_{ECD}$ in Expi293F cells (Thermo Fisher Scientific). As reported previously, the R470A mutation of LGI1 increases the yield of the LGI1–ADAM22$_{ECD}$ complex without affecting the higher-order assembly of LGI1–ADAM22$_{ECD}$ (*Yamagata et al., 2018*). The vectors for co-expression were the pEBMulti-Neo vector (Wako Chemicals) harboring the gene encoding human LGI1 (full length, residues 37–557; R470A) with the N-terminal Igκ signal sequence and that harboring the gene encoding human ADAM22$_{ECD}$, including the N-terminal prosequence (residues 1–729), both of which were reported previously (*Yamagata et al., 2018*). The culture media were loaded onto a Ni-NTA (QIAGEN) column pre-equilibrated with 20 mM Tris-HCl (pH 8.0) containing 300 mM NaCl. After the column was washed with 20 mM Tris-HCl (pH 8.0) containing 300 mM NaCl and 25 mM imidazole, the proteins were eluted with 20 mM Tris-HCl (pH 8.0) containing 300 mM NaCl and 250 mM imidazole. The eluted proteins were further purified by size-exclusion chromatography using Superdex200 (GE Healthcare) with 20 mM Tris-HCl (pH 7.5) buffer containing 50 mM NaCl. The purified proteins were concentrated to 0.25 g/L in Amicon Ultra-15 50,000 MWCO filter (Millipore), and glutaraldehyde was added at a final concentration of 0.1% (vol/vol). The sample was concentrated to 500 µL in Amicon Ultra-4 50,000 MWCO filter (Millipore) and purified by gel filtration chromatography using a Superose 6 10/300 GL (GE Healthcare) column with 20 mM Tris-HCl buffer (pH 7.5) containing 50 mM NaCl. One of the peak fractions was concentrated to 0.5 g/L by Amicon Ultra-0.5 50,000 MWCO filter (Millipore).

## Cryo-EM single-particle analysis

Cu grids (R1.2/1.3, 300 mesh, Quantifoil) with holey carbon films were hydrophilized using a JEC-3000FC Auto Fine Coater (JEOL) at 7 Pa, 10 mA, and 10 s. A 3 µL aliquot of 0.5 g/L protein solution was added to the grids using a Vitrobot Mark IV (Thermo Fisher Scientific) at a temperature of 8°C and 100% humidity. The grids were then immersed in liquid ethane and rapidly frozen under the following conditions: waiting time of 0 s, blotting time of 3 s, and blotting force of 10. Data collection was carried out on a CRYO ARM 300 transmission electron microscope (JEOL Ltd., Japan) operating at 300 kV, equipped with an Omega-type in-column energy filter (slit width 20 eV) and a Gatan K3 electron detector (operated in correlated doubling sample mode) at SPring-8 (Hyogo, Japan). A total of 7625 movies were automatically collected using SerialEM (*Mastronarde, 2005*). Hole centering was performed using yoneoLocr (*Yonekura et al., 2021*) integrated as a SerialEM macro. Movies were

collected using the beam-image shift method (5×5×1 matrices), at a target defocus range of −1.4 to −1.7 µm and a nominal magnification of ×60,000, corresponding to a calibrated pixel size of 0.752 Å/pixel. Each movie was recorded with an exposure time of 2.79627 s, subdivided into 60 frames with a total electron dose of 60.8046 e$^{-1}$ Å$^{-2}$.

All processing was performed in CryoSPARC v.4.1.2 or higher (*Punjani et al., 2017*). The collected micrographs were processed using patch motion correction and patch CTF estimation. Particles were picked from 7635 particle images using a template, yielding 3,336,559 particles. After extracting the particles from the micrographs and classifying them into 2D classes, about 60% of the particles based on the average 2D image of the protein were selected. An initial 3D reconstruction and heterogeneous refinement were performed on six classes, and one of these classes was further refined by nonuniform refinement to obtain a 2.78 Å resolution 3D map of the LGI1$_{LRR}$–LGI1*$_{EPTP}$–ADAM22$_{ECD}$ complex. Additionally, by performing ab initio reconstruction and heterogeneous refinement of one of the six classes into three subclasses and refining one of these subclasses further with nonuniform refinement (*Punjani et al., 2020*), a 3D map of the 3:3 LGI1–ADAM22$_{ECD}$ complex with a resolution of 3.79 Å was obtained. Overall resolution estimates correspond to a Fourier shell correlation of 0.143 using an optimized mask that is automatically determined after refinement. Local resolution maps were obtained using local resolution estimation. Movies were created using 3D VA (*Punjani and Fleet, 2021*) and 3D Flex (*Punjani and Fleet, 2023*).

## Model building

Model building was performed using the programs Coot (*Emsley and Cowtan, 2004*) and UCSF ChimeraX (*Meng et al., 2023*). The initial model of the LGI1$_{LRR}$–LGI1*$_{EPTP}$–ADAM22$_{ECD}$ complex was built using a part of the crystal structure of the 2:2 LGI1–ADAM22$_{ECD}$ complex (PDB 5Y31) (*Yamagata et al., 2018*) and fitted into the cryo-EM map. The initial model of the 3:3 LGI1–ADAM22$_{ECD}$ complex was built using the cryo-EM structure of the LGI1$_{LRR}$–LGI1*$_{EPTP}$–ADAM22$_{ECD}$ complex. Three LGI1$_{LRR}$–LGI1*$_{EPTP}$–ADAM22$_{ECD}$ complexes were fitted into the cryo-EM map, and LGI1$_{LRR}$ and LGI1$_{EPTP}$ were connected by modeling the linker region. The structure refinement was performed using the Phenix software package (*Adams et al., 2011*). All figures and movies were created using the program PyMOL (Schrödinger, LLC) and UCSF ChimeraX (*Meng et al., 2023*).

## HS-AFM observations

HS-AFM experiments were conducted using a custom-built HS-AFM operating in tapping mode (*Sumino et al., 2024*). Briefly, an optical beam deflection detector monitored the cantilever's (BL-AC10DS-A2, Olympus, Japan) deflection using a 780 nm, 0.8 mW infrared (IR) laser. The cantilever exhibited a spring constant of approximately 100 pN/nm, a resonant frequency near 400 kHz, and a quality factor around 2 in a liquid environment. The IR beam was directed onto the cantilever's back surface through a ×60 objective lens (CFI S Plan Fluor ELWD 60X, Nikon, Japan), and its reflection was detected by a two-segmented PIN photodiode (MPR-1, Graviton, Japan). The initial AFM tip had a triangular section resembling a bird's beak. To enhance spatial resolution, an amorphous carbon tip was grown on the bird beak tip via electron beam deposition using a scanning electron microscope (FE-SEM; Verious 5UC, Thermo Fisher Scientific, USA). The additional AFM tip was roughly 500 nm long, with an apex radius of approximately 1 nm after plasma etching with a plasma cleaner (Tergeo, PIC Scientific, USA). All HS-AFM images were obtained using the cantilever with the additional AFM tip. The cantilever's free oscillation amplitude was less than 1 nm, and during HS-AFM scanning, the set point amplitude was adjusted to roughly 90% of the free amplitude. To minimize interaction forces between the sample and the AFM tip, 'only trace imaging' (OTI) mode (*Fukuda and Ando, 2021*) was employed for HS-AFM.

HS-AFM observations of the LGI1–ADAM22$_{ECD}$ complex (not chemically cross-linked with glutaraldehyde) were conducted on AP-mica, prepared by treating the mica surface with 0.00005% (3-aminopropyl)triethoxysilane (APTES) (Shin-Etsu Chemical, Japan) in MilliQ water for 3 min. Samples at a concentration of 10 nM were added to the AP-mica surface after 3 min incubation of 3 µL. HS-AFM observations of the LGI1–ADAM22$_{ECD}$ complex were performed in 20 mM Tris-HCl (pH 7.4) buffer containing 150 mM NaCl. All HS-AFM experiments were performed at room temperature (24–26°C) and were independently repeated at least three times, consistently yielding similar results. For image processing, the HS-AFM images were processed using Fiji (ImageJ) software (NIH, USA) (*Schindelin*

*et al., 2012*). A mean filter with a radius of 0.5 pixels was utilized to lower noise levels in each image. The Template Matching and Slice Alignment plugin for ImageJ was used to correct for drift between sequential images.

## Simulation of AFM images

The BioAFMviewer software (*Amyot and Flechsig, 2020*) was utilized to validate the captured topographies of the LGI1–ADAM22$_{ECD}$ complex. The simulated scanning was based on the nonelastic collisions between a rigid cone-shaped tip model and the rigid van der Waals atomic model of the protein structure. Automatized fitting (*Amyot et al., 2022*) was employed to generate a simulated image that closely matched the HS-AFM target image (image correlation coefficients reported in each figure). In the simulation presented in *Figure 4*, the tip shape parameters were set to R=0.4 nm for the tip probe sphere radius and α=5.0° for the cone half angle.

## Cell lines

Expi293F cell lines were purchased from and authenticated by Thermo Fisher Scientific (A14527). The mycoplasma contamination test was confirmed to be negative.

## Acknowledgements

This research was partially supported by the Research Support Project for Life Science and Drug Discovery (Basis for Supporting Innovative Drug Discovery and Life Science Research [BINDS]) from AMED under Grant Number JP24ama121001. Preliminary cryo-EM experiments were performed at EM01CT/EM02CT with the approval of the Japan Synchrotron Radiation Research Institute (JASRI) (Proposal No. 2021B2542, 2022A2542, 2022B2543, and 2023A2543). This work was supported by JSPS/MEXT KAKENHI (19H03162 to AY, 23H00374 to MF, 18H03983 and 23H04070 to SF), Japan Agency for Medical Research and Development (JP24wm0625319 to YF and JP24ek0109649 to MF), Takeda Science Foundation (to SF), the World Premier International Research Center Initiative (WPI), MEXT, Japan (to MS).

## Additional information

### Funding

| Funder | Grant reference number | Author |
|---|---|---|
| Japan Society for the Promotion of Science | 19H03162 | Atsushi Yamagata |
| Japan Society for the Promotion of Science | 23H00374 | Masaki Fukata |
| Japan Society for the Promotion of Science | 18H03983 | Shuya Fukai |
| Japan Society for the Promotion of Science | 23H04070 | Shuya Fukai |
| Japan Agency for Medical Research and Development | JP24wm0625319 | Yuko Fukata |
| Japan Agency for Medical Research and Development | JP24ek0109649 | Masaki Fukata |
| Takeda Science Foundation | | Shuya Fukai |
| Ministry of Education, Culture, Sports, Science and Technology | WPI | Mikihiro Shibata |

| Funder | Grant reference number | Author |
|--------|------------------------|--------|

The funders had no role in study design, data collection and interpretation, or the decision to submit the work for publication.

## Author contributions

Takayuki Yamaguchi, Investigation, Writing – review and editing; Kei Okatsu, Data curation, Supervision, Investigation, Writing – review and editing; Masato Kubota, Formal analysis, Visualization; Ayuka Mitsumori, Formal analysis, Investigation, Visualization, Writing – original draft, Writing – review and editing; Atsushi Yamagata, Supervision, Funding acquisition, Investigation, Writing – review and editing; Yuko Fukata, Masaki Fukata, Conceptualization, Resources, Funding acquisition, Writing – review and editing; Mikihiro Shibata, Data curation, Formal analysis, Funding acquisition, Investigation, Visualization, Writing – original draft, Writing – review and editing; Shuya Fukai, Conceptualization, Data curation, Formal analysis, Supervision, Funding acquisition, Investigation, Visualization, Writing – original draft, Project administration, Writing – review and editing

## Author ORCIDs

Kei Okatsu ⓘ https://orcid.org/0000-0001-8949-3750
Atsushi Yamagata ⓘ https://orcid.org/0000-0001-9285-1256
Yuko Fukata ⓘ https://orcid.org/0000-0001-7724-8643
Masaki Fukata ⓘ https://orcid.org/0000-0001-5200-9806
Mikihiro Shibata ⓘ https://orcid.org/0000-0001-5041-3979
Shuya Fukai ⓘ https://orcid.org/0000-0002-1241-1443

Reviewer #1 (Public review): https://doi.org/10.7554/eLife.105918.3.sa1
Author response https://doi.org/10.7554/eLife.105918.3.sa2

## Additional files

### Supplementary files

MDAR checklist

### Data availability

The coordinates and maps of the $LGI1_{LRR}$–$LGI1*_{EPTP}$–$ADAM22_{ECD}$ complex and the 3:3 $LGI1$–$ADAM22_{ECD}$ complex have been deposited in the Protein Data Bank/Electron Microscopy Data Bank under the accession codes of 9KZC/EMD-62659 and 9KZT/EMD-62668, respectively.

The following datasets were generated:

| Author(s) | Year | Dataset title | Dataset URL | Database and Identifier |
|-----------|------|---------------|-------------|-------------------------|
| Yamaguchi T, Okatsu K, Kubota M, Mitsumori A, Yamagata A, Fukai S | 2025 | Cryo-EM structure of the LGI1 LRR-LGI1 EPTP-ADAM22 ECD complex | https://doi.org/10.2210/pdb9KZC/pdb | Worldwide Protein Data Bank, 10.2210/pdb9KZC/pdb |
| Yamaguchi T, Okatsu K, Kubota M, Mitsumori A, Yamagata A, Fukai S | 2025 | Cryo-EM structure of the LGI1 LRR-LGI1 EPTP-ADAM22 ECD complex | https://www.ebi.ac.uk/emdb/EMD-62659 | Electron Microscopy Data Bank, EMD-62659 |
| Yamaguchi T, Okatsu K, Kubota M, Mitsumori A, Yamagata A, Fukai S | 2025 | Cryo-EM structure of the 3:3 LGI1-ADAM22 complex | https://doi.org/10.2210/pdb9KZT/pdb | Worldwide Protein Data Bank, 10.2210/pdb9KZT/pdb |
| Yamaguchi T, Okatsu K, Kubota M, Mitsumori A, Yamagata A, Fukai S | 2025 | Cryo-EM structure of the 3:3 LGI1-ADAM22 complex | https://www.ebi.ac.uk/emdb/EMD-62668 | Electron Microscopy Data Bank, EMD-62668 |

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
