## [Editor Report · eLife Assessment]

In this **convincing** work by Yamaguchi et al. the cryo-EM structure of the heterohexameric 3:3 LGI1-ADAM22 complex is presented. The findings suggest that LGI1 can cluster ADAM22 in a trimeric fashion. The clustering of cell surface proteins is **important** in controlling signaling in the nervous system. This new version of the manuscript has been improved substantially and the figures have been enhanced and clarified.

---

## [Referee Report · Reviewer #1 (Public review)]

The structure of a heterohexameric 3:3 LGI1-ADAM22 complex is resolved by Yamaguchi et al. It reveals the intermolecular LGI1 interactions and its role in bringing three ADAM22 molecules together. This may be relevant for the clustering of axonal Kv1 channels and control over their density. While it is currently not clear if the heterohexameric 3:3 LGI1-ADAM22 complex has a physiological role, the detailed structural information presented here allows to pinpoint mutations or other strategies to probe the relevance of the 3:3 complex in future work.

The experimental work is done to a high standard, and all my comments have been addressed. This new version of the manuscript has been improved substantially, and the figures have been enhanced and clarified.

---

## [Author Response]

The following is the authors’ response to the original reviews

**Public Reviews:**

**Reviewer #1 (Public review):**
(1) A previously determined 2:2 heterodimeric complex of LGI1-ADAM22 was suggested to play a role in trans interactions. Could the authors discuss if the heterohexameric 3:3 LGI1-ADAM22 is more likely to represent a cis complex or a trans complex, or if both are possible?

We noticed that there was no obvious structural feature strongly suggesting that the heterohexameric 3:3 LGI1-ADAM22 is more likely to represent a *cis* complex or a *trans* complex. Both are possible at the synapse (and similarly, for LGI3-ADAM23 at the jaxtaparanode of myelinated axons). Therefore, we revised the Introduction and Discussion sections as follows:

Introduction: (about potential structural mechanisms of the 3:3 complex)

“Similarly to the 2:2 complex, the 3:3 complex might serve as an extracellular scaffold to stabilize Kv1 channels or AMPA receptors in a *trans*-synaptic fashion. In addition, the 3:3 assembly in a *cis* fashion on the same membrane might regulate the accumulation of Kv1 channel complexes at axon initial segment. However, no clear evidence to prove these potential mechanistic roles of the 3:3 assembly has been provided, and the three-dimensional structure of the 3:3 complex has not yet been determined.”

Discussion: (about a role of the LGI3–ADAM23 complex at the jaxtaparanode of myelinated axons)

“In this context, as discussed in (30), either or both of the 2:2 and 3:3 complexes might be formed in a *trans* fashion at the juxtaparanode of myelinated axons and bridge the axon and the innermost myelin membrane. Alternatively, the 3:3 complex formed in a *cis* fashion might positively regulate the clustering of the axonal Kv channels at the juxtaparanode, possibly in a similar manner at the axon initial segment.”

*Ref. 30: Y. Miyazaki *et al.*, Oligodendrocyte-derived LGI3 and its receptor ADAM23 organize juxtaparanodal Kv1 channel clustering for short-term synaptic plasticity. Cell Rep 43, 113634 (2024).

(2) It is not entirely clear to me if the LGI1-ADAM22 complex is also crosslinked in the HS-AFM experiments. Could this be more clearly indicated? In addition, if this is the case, could an explanation be given about how the complex can still dissociate?

Thank you for the constructive suggestions. A non-crosslinked 3:3 LGI-ADAM22 complex was used for HS-AFM observations. To clarify the sample used for HS-AFM, we have modified the text as follows.

P.8 “Dynamics of the LGI1‒ADAM22 higher-order complex observed by HS-AFM

HS-AFM images of gel filtration chromatography fractions containing the 3:3 LGI1-ADAM22_ECD_ complex (not chemically crosslinked with glutaraldehyde) predominantly…”

P.10 Materials and methods

“HS-AFM observations of the LGI1–ADAM22_ECD_ complex (not chemically crosslinked with glutaraldehyde) were conducted on AP-mica,…”

(3) The LGI1 and ADAM22 are of similar size. To me, this complicates the interpretation of dissociation of the complex in the HS-AFM data. How is the overinterpretation of this data prevented? In other words, what confidence do the authors have in the dissociation steps in the HS-AFM data?

Our criteria for assigning HS-AFM images to the 3:3 LGI1–ADAM22_ECD_ complex were based on a comparison of the simulated AFM image of the 3:3 complex obtained by cryo-EM. The automatized fitting process (42) identifies the optimal orientation of cryo-EM images that closely matches the HS-AFM image. In the present study, the concordance coefficient (CC) reached 0.8, indicating that the protein orientation in HS-AFM images of the 3:3 complex was objectively satisfactory.

Regarding the dissociation step of ADAM22 from the 3:3 complex, we carefully analyzed the HS-AFM videos frame by frame and observed that the protrusion corresponding to ADAM22 in the 3:3 complex disappeared at a specific frame (4.5 s in the third molecule in Movie S1). The dissociation steps of ADAM22 were further confirmed by integrating multiple independent HS-AFM experiments and observations. Thus, although HS-AFM images alone cannot determine the orientation of LGI1 and ADAM22 in the 3:3 complex, the comparison of cryo-EM images with simulated AFM images enables objective assignment and orientation of proteins in the 3:3 complex through automated fitting.

*Ref. 42: R. Amyot *et al.*, Flechsig, Simulation atomic force microscopy for atomic reconstruction of biomolecular structures from resolution-limited experimental images. PLoS Comput Biol 18, e1009970 (2022).

(4) What is the "LGI1 collapse" mentioned in Figure 4c?

Thank you for the constructive suggestions. The term “LGI1 collapse” was intended the dissociation of LGI1 from the 3:3 complex. To avoid confusion, we have revised it to “LGI1 release”.

(5) Am I correct that the structure indicates that the trimerization is entirely organized by LGI1? This would suggest LGI1 trimerizes on its own. Can this be discussed? Has this been observed?

Yes. The present cryo-EM structure of the 3:3 complex indicates that the trimerization can be entirely organized by LGI1. In addition, during the HS-AFM imaging, the triangle shape seems to be maintained even if one ADAM22_ECD_ molecule is released. These findings suggest the possibility that LGI1 could trimerize on its own although this possibility could not be tested due to the difficulty in the expression of the full-length LGI1 alone for biophysical analysis in our hands. On the other hand, considering the dynamic property of the 3:3 complex and spatial alignment of LGI1LRR and ADAM22, we cannot exclude the possibility that ADAM22 could act as a platform to facilitate the intermolecular interaction between LGI1_LRR_ and LGI1*_EPTP_ for the trimerization of LGI1. This discussion was added in the first paragraph of the subsection "Dynamics of the LGI1–ADAM22 higher-order complex by HS-AFM".

(6) C3 symmetry was not applied in the cryo-EM reconstruction of the heterohexameric 3:3 LGI1-ADAM22 complex. How much is the complex deviating from C3 symmetry? What interactions stabilize the specific trimeric conformation reconstructed here, compared to other trimeric conformations?

According to this comment, we compared the non-symmetric, present cryo-EM structure to the previously calculated _C_3 symmetry-restrained structure based on small-angle X-ray scattering analysis and the _C_3 symmetric structure generated by AlphaFold3. Their differences in the domain or protomer configuration are illustrated in Fig. S9.

We did not find interactions that could obviously stabilize the specific trimeric conformation but the closure motion of LGI1_LRR_ (relative to LGI1_EPTP_) in chain F appears to locate it in close proximity to LGI1LRR in chain D to make the triangular assembly slightly more compact. This (partly) compact configuration might stabilize the non-symmetric trimeric configuration observed in the cryo-EM structure. This was described in the last sentence in the subsection "Cryo-EM structure of the 3:3 LGI1– ADAM22_ECD_ complex".

**Reviewer #2 (public review):**
The functional significance of these two complexes in the context of synapse remains speculative.

To assess the functional significance of the 3:3 complex, we spent time and effort designing mutations that solely inhibit the 3:3 assembly but failed to find such mutations. In this paper, we just focused on structural characterization of the 3:3 complex.

Additionally, the structural presentations in Figures 1-3 (especially Figures 2-3) lack the clarity needed for general readers to fully understand the authors' key points. Enhancing the quality of these visual representations would greatly improve accessibility and comprehension.

We made an effort to improve Figures 1-3 accordingly. Specifically, we revised them based on the strategy suggested in the Editorial comment regarding this reviewer's comment.

**Editorial comments:**
We noticed that in the reconstruction of the 3:3 complex, which is claimed to be at 3.8A resolution, beta-strands are not separated in the map and local resolution estimates vary from 6-10A. Please clarify.

We revised Fig. S8 to show the local resolution and volume quality, which correspond to nominal resolution of 3.8 Å, estimated from gold-standard FSC.

**Reviewer #1 (Recommendations for the authors):**
(1) PDB validation reports should be presented to allow further validation

The PDB validation reports were attached to the revised manuscript (uploaded as "related manuscript file").

(2) In Figure 4, models below the AFM figures are difficult to see because of the light coloring. In addition, in panel c, the orientation of some of the parts of the models below the 19.2 and 34.5 s. panels do not seem to correlate with the AFM figures. Could the models be adjusted so that they represent the data better?

Thank you for the constructive suggestions. According to the Reviewer’s comments, we have revised the AFM figures (Fig. 4).

(3) References are sometimes missing for important statements. Please check throughout.Some examples:P3, "it has been suggested that the 3:3 complex regulates the density of synaptic molecules such as scaffolding proteins and synaptic vesicles".P3. "Furthermore, LGI1 forms a complex with the voltage-gated potassium channel (VGKC) through ADAM22/23".

According to this comment, we rewrote the description about potential physiological roles of the 3:3 complex and added references as follows:

"Similarly to the 2:2 complex, the 3:3 complex might serve as an extracellular scaffold to stabilize Kv1 channels or AMPA receptors in a *trans*-synaptic fashion (9, 17, 19). In addition, the 3:3 assembly in a *cis* fashion on the same membrane might regulate the accumulation of Kv1 channel complexes at axon initial segment (18, 20). However, no clear evidence to prove these potential mechanistic roles of the 3:3 assembly has been provided, and the three-dimensional structure of the 3:3 complex has not yet been determined."

We also added references to the following sentences:

p.2, (the last sentence in the first paragraph of the Introduction) “Additionally, some epilepsy-related mutations have been identified in genes encoding non-ion channel proteins such as *LGI1* (4-7).”

p.3, ln 4-5, “The metalloprotease-like domain interacts with the EPTP domain of LGI1 in the extracellular space (11, 14).”

p.3, ln 9-10, “Furthermore, LGI1 forms a complex with the voltage-gated potassium channel (VGKC) through ADAM22/23 (9, 17, 18)”

p.3, ln 20-22, “The results revealed the structural basis of the interaction between the EPTP domain of one LGI1 and the LRR domain of the other LGI1, as well as the interaction between the EPTP domain of LGI1 and the metalloproteinase-like domain of ADAM22 (14)”

(4) S5 for clarity please add an overview of the complex highlighting where the different parts shown in the panels are located.

Fig. S5 was modified accordingly. Every panel showing a zoom-up view was indicated by a box in an overview of the complex.

(5) S7 a+b, also here add models for the structures to indicate which parts are shown.Could labels be added to highlight important parts?

We added an overview of the complex with boxes that indicate the parts shown as the panels, according to this comment. We also added labels to highlight residues that are important for the LGI1_EPTP_–ADAM22_ECD_ interaction in the panel showing the LGI1_EPTP_–ADAM22_ECD_ interface.

(6) S7c also shows the cartoon of the structure. How is it possible that the local resolution is not much higher than 6 Å? The overall resolution was 3.8 Å? This looks like a figure of the density plotted at a low level, and not as stated a "surface representation". Could an extra panel be shown of the density plotted at a higher level? Also, please add Å to the legend in this figure.

Local resolution maps of the 3:3 LGI1-ADAM22_ECD_ complex were shown as Fig. S8 in the revised manuscript. According to this comment, the distribution of the resolution was plotted onto the density at high (0.06) and low (0.03) levels. "Å" was added to the legend in the figure.

**Reviewer #2 (Recommendations for the authors):**
(1) The study was conducted using the ectodomain (ECD) of ADAM22. It remains unclear whether the 3:3 complex could form if the transmembrane domain (TMD) of ADAM22 were included. In other words, it is difficult to assess whether the observed 3:3 complex represents plausible cis interactions.

As mentioned in our reply to the first comment from Reviewer #1, we noticed that there was no obvious structural feature strongly suggesting that the heterohexameric 3:3 LGI1–ADAM22 is more likely to represent a *cis* complex or a *trans* complex. Both are possible at the synapse (and similarly, for LGI3–ADAM23 at the jaxtaparanode of myelinated axons). Therefore, we revised the Introduction and Discussion sections as follows:

Introduction: (about potential structural mechanisms of the 3:3 complex)

“Similarly to the 2:2 complex, the 3:3 complex might serve as an extracellular scaffold to stabilize Kv1 channels or AMPA receptors in a *trans*-synaptic fashion. In addition, the 3:3 assembly in a *cis* fashion on the same membrane might regulate the accumulation of Kv1 channel complexes at axon initial segment. However, no clear evidence to prove these potential mechanistic roles of the 3:3 assembly has been provided, and the three-dimensional structure of the 3:3 complex has not yet been determined.”

Discussion: (about a role of the LGI3–ADAM23 complex at the jaxtaparanode of myelinated axons)

“In this context, as discussed in (30), either or both of the 2:2 and 3:3 complexes might be formed in a *trans* fashion at the juxtaparanode of myelinated axons and bridge the axon and the innermost myelin membrane. Alternatively, the 3:3 complex formed in a *cis* fashion might positively regulate the clustering of the axonal Kv channels at the juxtaparanode, possibly in a similar manner at the axon initial segment.”

*Ref. 30: Y. Miyazaki *et al.*, Oligodendrocyte-derived LGI3 and its receptor ADAM23 organize juxtaparanodal Kv1 channel clustering for short-term synaptic plasticity. Cell Rep 43, 113634 (2024).

(2) Page 2, line 1: "...caused by genetic mutations." - Specify the mutations involved. Which genes are mutated? Providing this information would enhance clarity and context.

According to this comment, we rephrased the sentence as follows:

"LGI1 is linked to epilepsy, a neurological disorder that can be caused by genetic mutations of genes regulating neuronal excitability (e.g., voltage- or ligand-gated ion channels)."

(3) The experimental strategy and data for both cryo-EM and HS-AFM are of high quality. However, improvements are needed in the cryo-EM/structural figures to enhance clarity. Structural components should be labeled, and the protein interfaces should be identified within the overall complex figures in Figures 2 and 3, as the current presentation is challenging for general readers to follow. For example, in Figure 2, panel a would benefit from clear labeling to indicate the locations of ADAM22 and LGI1. Panels b and c lack context unless the authors specify which interface corresponds to panel a. Additionally, panels e and f are unlabelled, making it difficult to interpret the figures. Improved annotations and descriptions would significantly enhance figure accessibility and comprehension.

Thank you for the constructive suggestion for enhancing accessibility and comprehension of cryo-EM/structural figures. According to this comment, we labeled structural components and indicated the protein interfaces as boxes in the overall complex figures in Figures 2 and 3. Further, in Figure 2, the locations that panels b and c show were indicated as two boxes in the close-up view in panel a.